# High Resolution 3D Model of Heritage Landscapes Using UAS LiDAR: The Tajos de Alhama de Granada, Spain

**María del Carmen Vílchez-Lara \***, **Jorge Gabriel Molinero-Sánchez**, **Concepción Rodríguez-Moreno, Antonio Jesús Gómez-Blanco** and **Juan Francisco Reinoso-Gordo**

Department of Architectural and Engineering Graphic Expression, University of Granada, 18071 Granada, Spain; jmolinero@ugr.es (J.G.M.-S.); crodriguezmoreno@ugr.es (C.R.-M.); agomezb@ugr.es (A.J.G.-B.); jreinoso@ugr.es (J.F.R.-G.)
\*  Correspondence: mariacarmenvl@ugr.es

**Abstract:** The Tajos de Alhama de Granada, which since ancient times have inspired and surprised locals and strangers, especially foreign travelers, constituted a unique landscape, cultural and ethnological heritage of Spain, linked to water and its old flour mills. And, they are currently at serious risk of degradation. The aim of this research is to obtain a high-resolution 3D model capable of documenting this historical heritage environment with a high level of detail, using a methodology that includes small light weight LiDAR (Light Detection and Ranging) system for UAS (Unmanned Aircraft System). The model obtained should serve, on the one hand, as a valuable tool for knowledge and analysis of all the elements (river, lake, ditches, dams, mills, aqueducts, and paths) that made up this place, registered as a picturesque landscape for its extraordinary beauty and uniqueness, and on the other hand, as a basis for the development of rehabilitation and architectural restoration projects that would have to be undertaken to preserve this cultural and landscape legacy.

**Keywords:** landscape; cultural heritage; LiDAR; UAS; photogrammetry; 3D model; point cloud

## 1. Introduction

The main objective of this research is to obtain a high-resolution model that allows the identification, with a high level of detail, of each element that makes up this heritage landscape, through a structured methodology that allows the combined use of UAS LI-DAR, understanding UAS according to the definition by the International Civil Aviation Organization (ICAO): an aircraft and its associated elements are operated with no pilot on board [1]. A point cloud obtained by UAS has already proved to be useful to capture heritage elements, both obtained by photogrammetry [2] or by laser scanning. [3,4]. Airborne LiDAR technology has proved to be useful in archaeological and heritage discoveries where there were difficult to access wooded areas, thanks to tree penetration [5,6]. Given the limited geographical scope of our research, we decided to use UAS LiDAR, which has already proved to be sufficiently accurate in studies closed to ours [7,8]. However, due to the existence of large vertical walls in Los Tajos de Alhama, we believe it is necessary to use a LiDAR that allows 360° rotation of the laser beams so fewer passes are necessary, and consequently reduces recording errors. The LiDAR technology is also necessary in the Tajos de Alhama because there is a lot of vegetation in the river area. The LiDAR has the ability to penetrate and discern the ground under the vegetation which is necessary to produce a reliable 3D terrain model. There are studies using this type of LiDAR that show adequate accuracy [9–11].

We consider this study carried out is of great importance because, although the heritage environment of the Tajos de Alhama was cataloged at the highest level of heritage protection granted by Spain, as an Asset of Cultural Interest and a National Monument, it currently lacks a detailed survey that allows it to be the basis of a deep knowledge of all

the parts that composed it and serves for any future recovery action, whether at a global or territorial level, for any of its elements.

The Spanish word "tajos" translates in English as gorges, is used to define the rocky and steep walls, practically vertical, of a deep and narrow valley through which a river runs, in our case called Alhama or Marchán.

The Tajos de Alhama have a length of three kilometers and two hundred meters between the artificial Pantaneta Lake, cataloged within the inventory of wetlands of Andalusia [12], and the last turn that the river makes when leaving the town of Alhama de Granada. A fundamental part of this extraordinary landscape is the historic hydraulic mills, located at the bottom of the gorges that delimit the urban edge on two of its sides. This industrial heritage is made up of five flour mills, four of which are completely abandoned and in progressive deterioration. The one called La Purísima is converted into a museum, although it is in a poor state of conservation in some of its parts, such as the structure of some floors and the roof.

The importance of the Tajos de Alhama is recognized at the level of protection and cataloging with different categories of protection: Asset of Cultural Interest linked to the Historical Complex of Alhama [13], Natural Monument of Andalusia [14], Picturesque Site [15], and Outstanding Landscape [16]. However, all these recognitions and inscriptions in National and Andalusian Heritage catalogs have not prevented the progressive state of degradation of its hydraulic flour mills and its closest surroundings, within the Tajos.

Located in the west of the province of Granada, its orography was decisive in the choice of the place where the first Muslims settled in the 9th century, a time when it received its current name, which came from the Arabic Al Hamma, which meant The Bath. It is from then on the city begins to be recognized as one of the most important in Andalusia. The route that linked the coast of Malaga with Granada passed through it.

The first human footprints that were found in the Tajos were prehistoric, from the Neolithic, in the missing Molinos Cave located in the El Cañón area, next to the urban center and one of the Tajos mills, that of San Francisco [17]. Thanks to the writings of the Roman Pliny the Elder and the Greek Ptolemy, we know a Roman city existed, although it was difficult to determine if it was located in the same place as the ancient Arab neighborhood of Alhama, as Ceán Bermúdez asserted in the 19th century [18], or next to the famous baths of Roman origin [19]. At that time, this city was known by the name of Artigi Iuliensis or Artigis and appeared represented on the Renaissance maps of Roman Hispania in the different versions of the *Geographia*, based on the longitude and latitude coordinates given in the 2nd century by Ptolemy to locate the main cities and geographical features [20]. In the 16th century, geographers such as Mercator [21] continued to reinterpret Ptolemy's treatise and produced increasingly precise maps of the Roman peninsula. A truly significant evolution was found in the middle of the 17th century in Briet's *Hispaniae Veteris*, Figure 1 [22], in which Artigis was located at the foot of the Penibética mountain range (Illipula). On this map, we can see the Marchán River passes through Artigis and flows into the Genil River, a tributary of the Gualdalquivir River (Baetis fl.).

The splendor of Alhama reached its culmination in the Nasrid era, when it became a strongly walled city, helped on three of its sides by the height of its vertical gorges. Its Arab end came in 1482 with the assault of the city by Christian troops.

The Tajos appeared represented for the first time in the view of the city and its surroundings drawn by Anton van der Wyngaerde in 1567, Figure 2. At that time the walls were still preserved, larger in the western area of the city (in the right side of the image), because this part lacked the large natural and vertical ditch that the gorges represented when bordering the city on its north, east, and south sides.

The cartographer and military engineer Pedro Texeira placed Alhama on his 1634 map [23] among the most important cities belonging to the Kingdom of Granada, Figure 3. In the 1696 map of the Italian geographer Giacomo Cantelli [24], Alhama appears at the feet of the Marchán River, which runs parallel to the Cacín River, until it flows into the Genil River,

Figure 4a. The ecclesiastical map, drawn in 1712 by Panicale and Montecalerio [25] of the province of Andalusia, included the cities where there were houses and convents of the Capuchin Franciscans, among them, Alhama, Figure 4b.

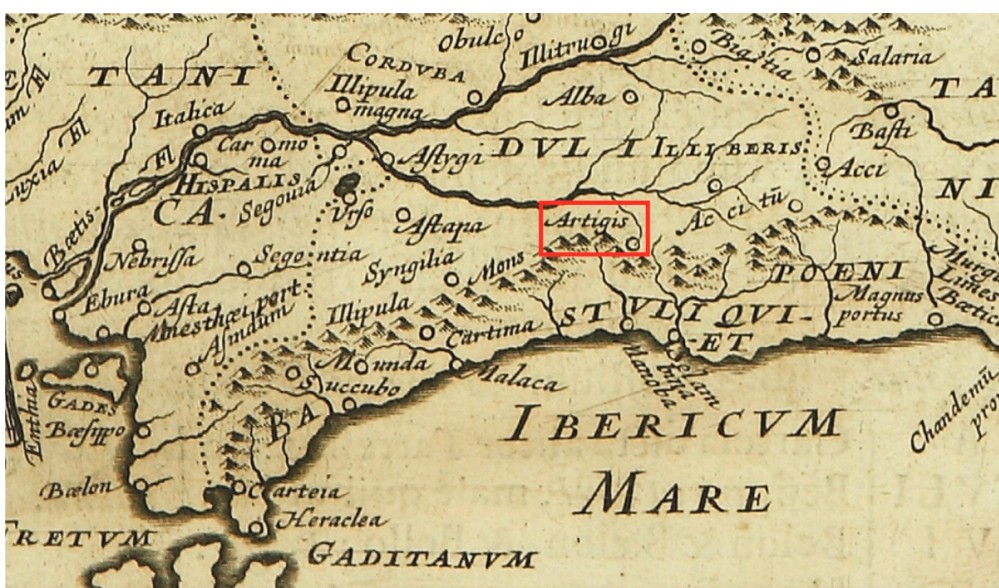

**Figure 1.** Artigis located on the *Hispaniae Veteris* map (Briet, 1649). Digital Historical Map of Extremadura.

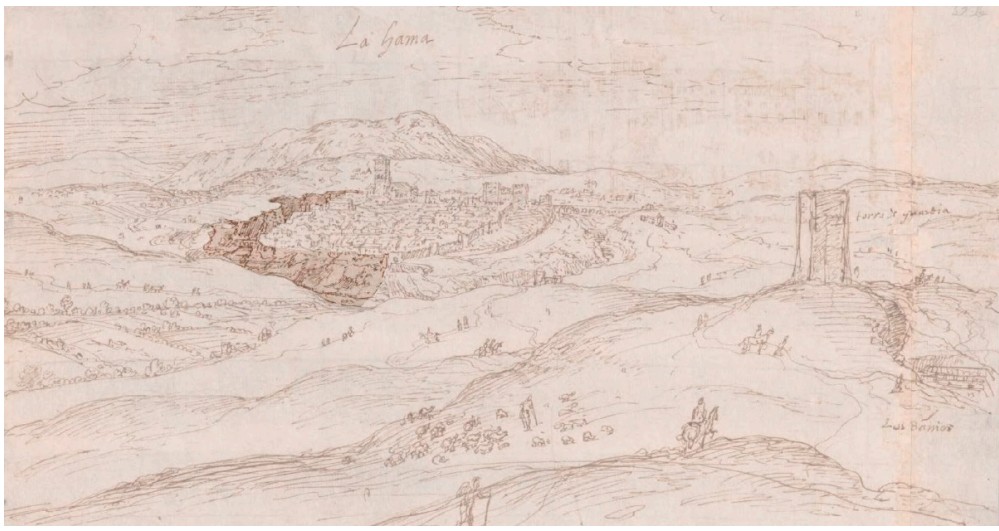

**Figure 2.** The Tajos highlighted in the view of Alhama (Wyngaerde, 1567). Vienna National Library.

Since the end of the 18th century, some travelers have left their written testimony about the landscape of the Tajos and its relationship with the city of Alhama [26]. In 1787, the geologist Townsend described them as a large precipice formed by high rock walls through which a river run at the bottom, both through numerous sonorous waterfalls and silent currents that slided gently over the terrain. He praises the beauty of this landscape that surrounds Alhama on all sides except for the west where a magnificent castle is located, at that time already in ruins. The Spanish Antonio Ponz was impressed by the very high almost perpendicular gorge that he calculated to be more than two hundred yards deep (around 170 m), discovering in its depth beauty in the succession of groves, mills, orchards, etc. Already in the 19th century we find the traces left by travelers like the North American Robert Semple, who even by starlight found the beauty of the rocky gorges and their romantic precipices that rose about 300 feet high above a valley to be immeasurable, along

which ran a stream of water bordered by trees and where several mills were located. The English travelers William Jacob, Charles Scott, and Richard Ford also wrote about the picturesque water mills at the bottom of the valley and the location of the city of Alhama surrounded by the Tajos. They also highlighted the hanging houses of Alhama on the edge of the cliffs. French travelers arrived in the city such as Théophile Gautier, who defined the Tajos as enormous cracks in the ground and to whom a beautiful lithograph made from one of the town's viewpoints was attributed, Figure 5, and Nicolas Chapuy, who made another lithograph of the Tajos with the city in the background. Also, a woman, Lady Tenison, left written testimony of her passage through the Tajos de Alhama, with high perpendicular rock walls, being impressed by the houses built on the edge of the precipice. Unfortunately, some of which came off in the great earthquake of 1884.

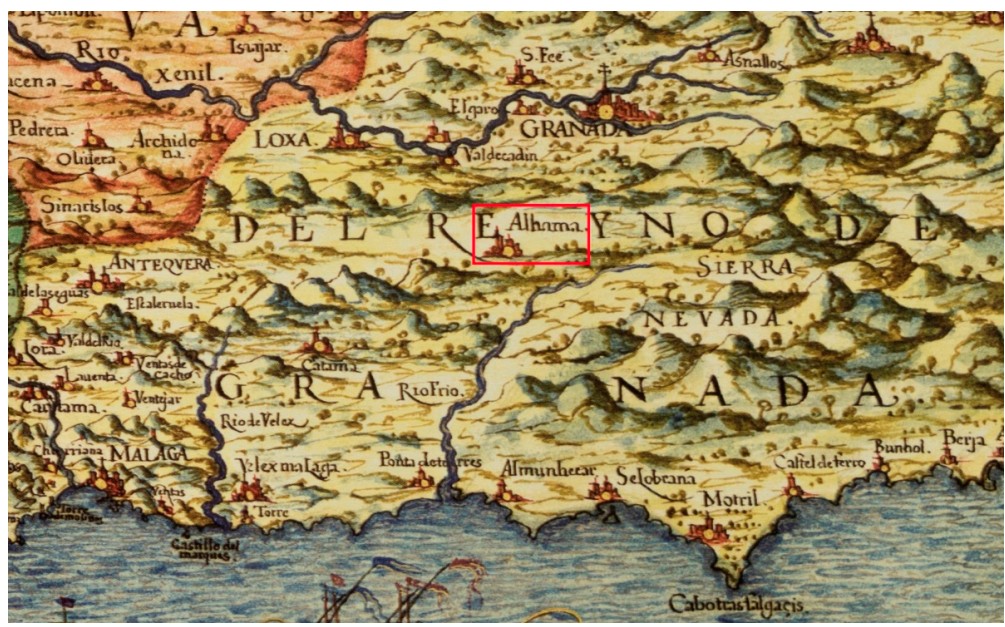

**Figure 3.** Alhama located on the Reyno de Andaluzia map (Texeira, 1634). Digital Historical Map of Extremadura.

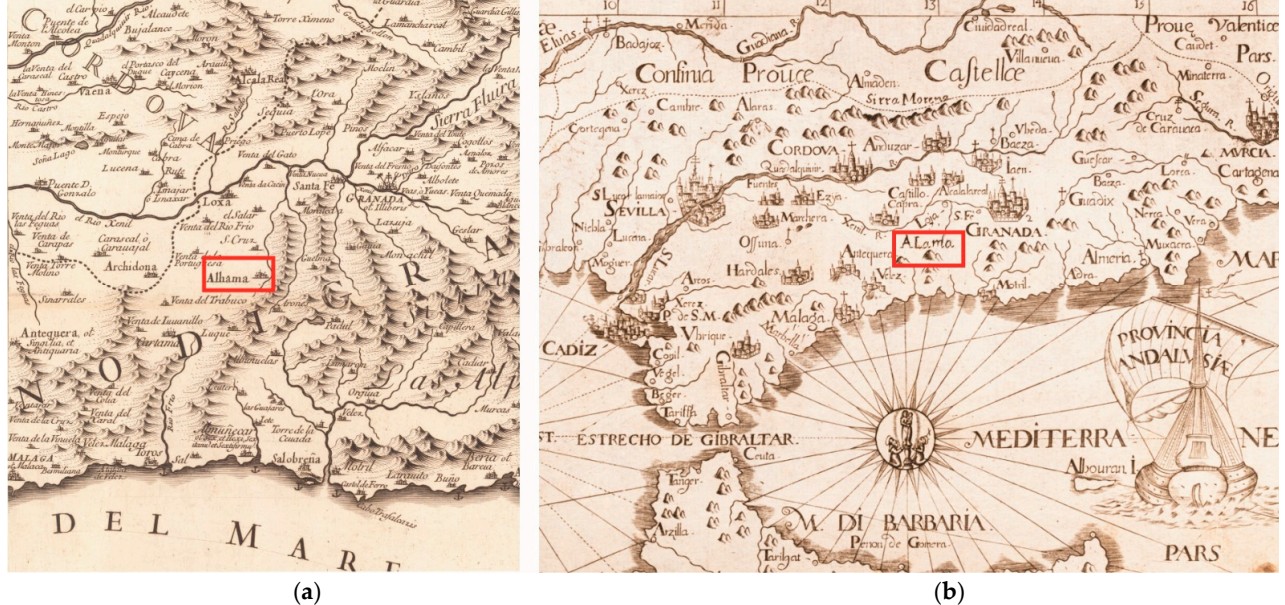

(**a**)  (**b**)

**Figure 4.** Alhama on the maps of (**a**) 1696 (Cantelli) and (**b**) 1712 (Panicale and Montecalerio). National Geographic Institute.

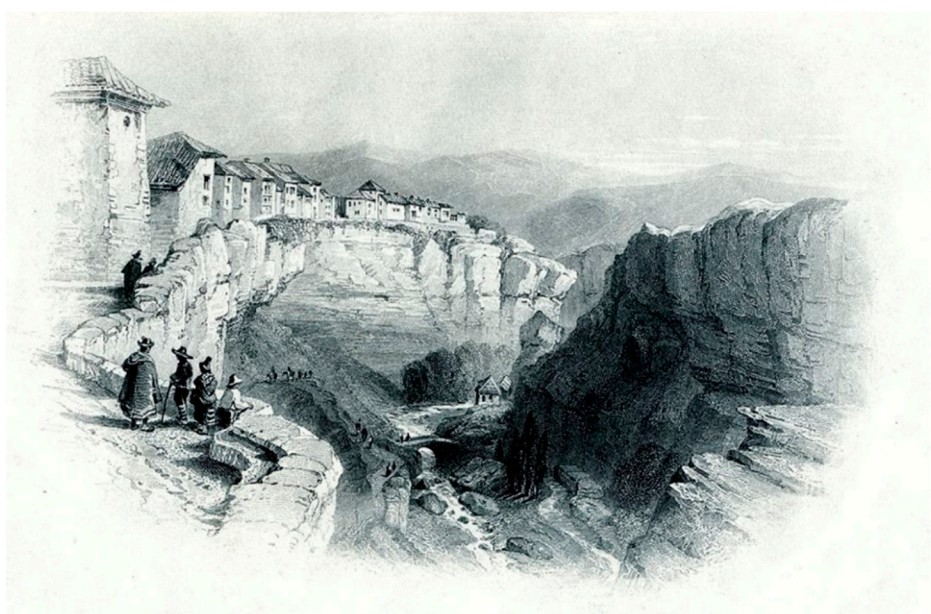

**Figure 5.** Lithograph of Alhama (attributed to Gautier, 1840).

## 2. Materials and Methods

### 2.1. Study Area

The Tajos de Alhama are located to the west of the province of Granada, almost on the border with the province of Málaga, Figure 6, within a territory that is made up of rounded hills, with the Penibética Mountains as a backdrop. The Alhama and Cacín rivers, which flow almost parallel, constitute the backbone of this region, both originating in the Almijara Mountain and flowing into the Genil River, in the Granada Meadow.

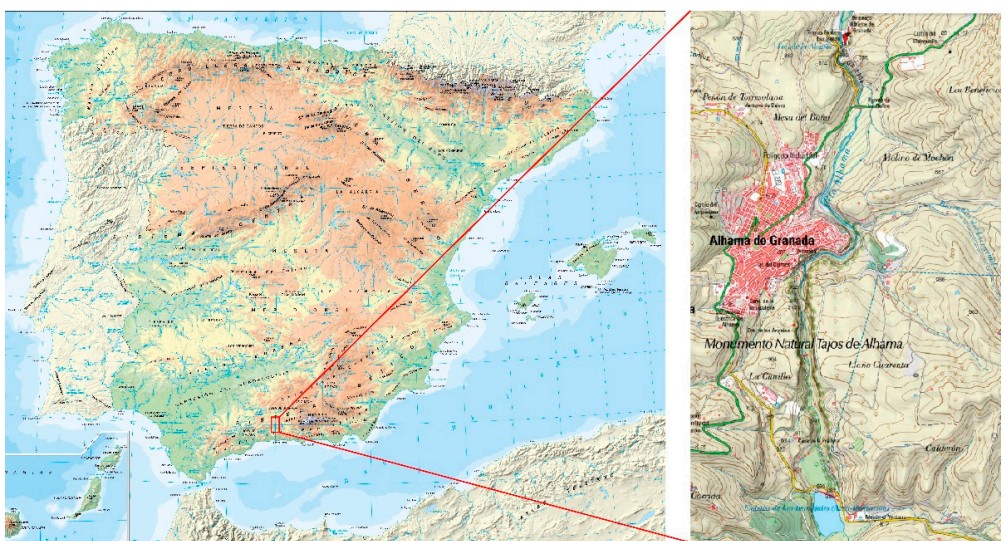

**Figure 6.** Situation of the study area in Spain and in the region. National Geographic Institute.

The Tajos de Alhama were declared a Natural Monument by Decree on 30 December 2011 and included in the General Catalog of Andalusian Historical Heritage. Since ancient times, they have formed a cultural landscape of great beauty, generated by, and linked to the water of the river that ran along its bottom from south to north. The tour of this area began at the artificial Pantaneta Lake, created in the middle of the 20th century to regulate the flow of water while accumulating it for periods of drought. But, at the end of the 1960s it was decided to transfer water from this small wetland and reservoir to the artificial Bermejales Lake, through a channel that ran underground and which we could see

in Figure 6 in dashed blue lines. Unfortunately, for the Tajos area this had negative effects in terms of the quantity and quality of the water that flew through the river from then on, causing, among others, the progressive closure of the old flour mills.

The water landscape of the Tajos is not only reduced to the Marchán River, but also ravines and irrigation ditches are running through it. The most notable in terms of importance and size was the so-called Caz del Molino that fed both the small orchards and the five flour mills, Figures 7 and 8. This irrigation ditch, which began its journey at the Los Angeles Dam, took advantage of the downward slope of the terrain so the water could obtain the necessary force to move the horizontal wheel hydraulic mechanisms of the mills. It ran parallel to the river until it flowed into it after moving the last mill, that of the Nuestra Señora del Carmen Mill (Flour Factory). Nowadays, it only carries water until just before entering the first mill, that of the San Francisco Mill (Flour Factory), at which point a gate prevents it from continuing its old route, returning the water to the river through a spillway that is located in said point.

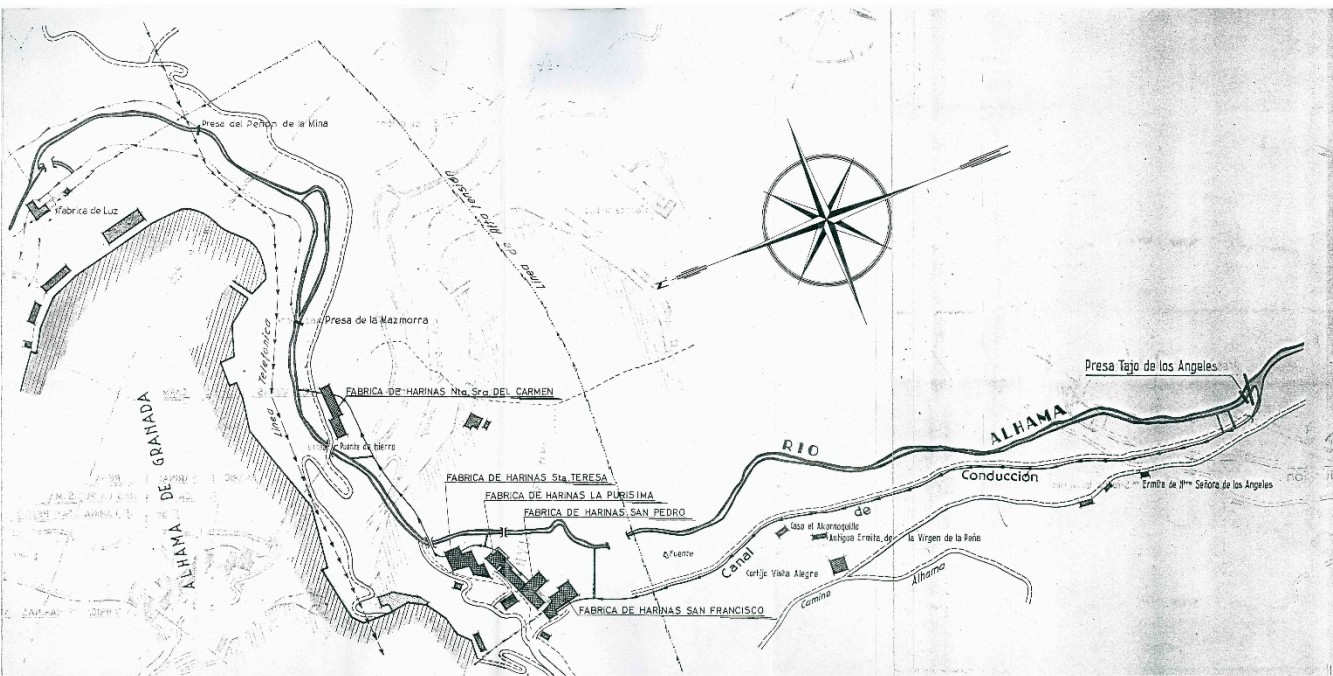

**Figure 7.** Industrial use of water from the Alhama River. Location of the mills (old flour factories), the Alhama River, and the Caz del Molino irrigation ditch. Plan of the bottom of the Tajos from the Los Ángeles Dam to the end of them in Alhama de Granada. https://www.fcalapurisima.com/p/el-agua-su-uso-industrial.html (accessed on 6 December 2022).

### 2.2. LiDAR System for UAS

LiDAR technology allows us to calculate the distance of far objects using laser pulses. Its topographic use to measure terrain elevations began in the 1980s airborne in manned aircraft [27]. The development in recent years of UAS (Unmanned Aircraft System), more commonly known as Drones, has made it possible to board small and light LiDAR equipment on UAS, more affordable and practical than helicopters or airplanes, in cases where the terrain to survey was not too extensive nor was it necessary to fly at high altitude. LiDAR systems must be assisted by a GNSS, to measure the exact position of the sensor, and an IMU (Inertial Measurement Unit) [28], to measure the exact orientation or inclination of the sensor.

The LiDAR equipment used was the Velodyne Puck Hi-Res (Alameda, CA, USA) with 16 beams, 2 returns and a positional uncertainty between 3 and 5 cm, according to the manufacturer's specifications. The vertical field of view has a 20° range (10° above and below its own horizon), its vertical resolution is 1.33° and its scanning capacity 360°

along its whole horizon, so it can be considered its horizontal field of view is 360°. This 360° capability motivated our choice rather than other equipment requiring less additional work. The flour mills environment is characterized by large slopes making up almost vertical walls and its LiDAR survey can be guaranteed setting the Velodyne in horizontally position and sweeping the terrain in the form of transverse profiles. Its horizontal angular resolution was 0.1° and the wavelength used was 903 nm. On the other hand, we used the UAS DJI's Matrice 300 RTK to carry the LiDAR with a flight time capacity up to 55 min (Table 1) (Figure 9a).

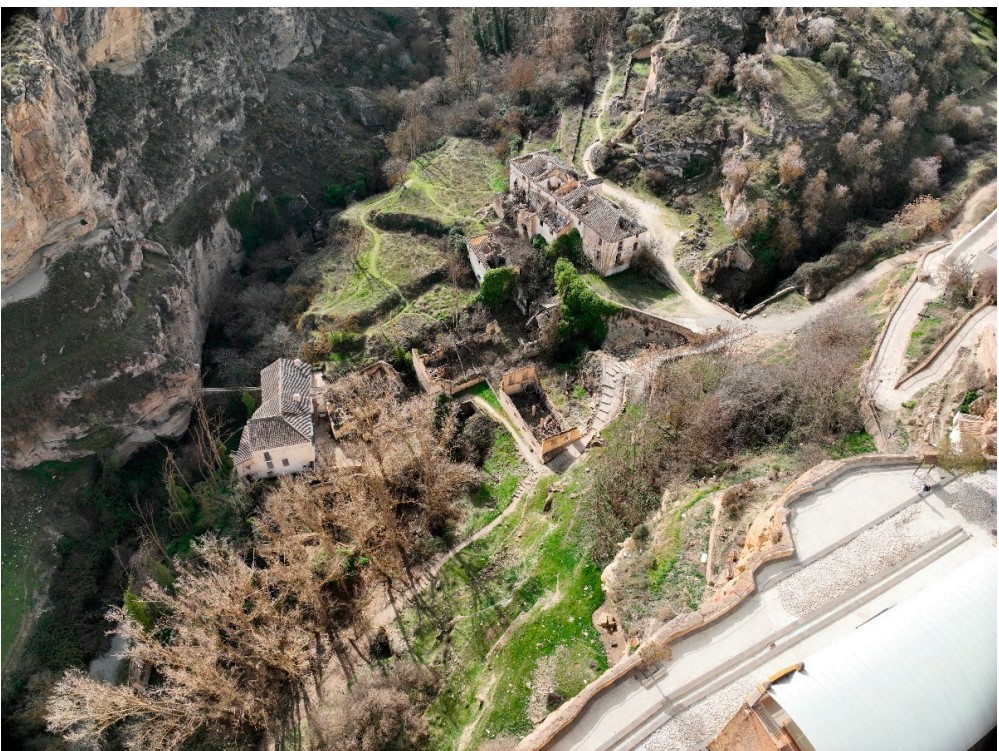

**Figure 8.** Mills at the bottom of the Tajos. Photography taken with DJI Mavic 3 pro.

**Table 1.** LiDAR on board UAS sensor features.

| LiDAR | Features |
|---|---|
| Real Time Kinematic (RTK) module | GNSS capabilities: (GPS: L1C/A; L2C/L2P; BDS: B1I B2I; GLO: G1 G2; GAL: E1 E5b; QZSS: L1 L2) Fixed RTK uncertainty: Horizontal: 1 cm + 1 ppm; vertical: 1.5 cm + 1 ppm |
| Sensor LiDAR | Velodyne Puck Hi-Res. 16 beams, 2 returns and 3 to 5 cm positional uncertainty |
| FOV Vertical | −10° to +10° Resolution 1.33° |
| FOV Horizontal | 360° Resolution 0.1° |
| Wave length | 903 nm |
| IMU | Honeywell high-resolution 0.08/0.03 degrees uncertainty |

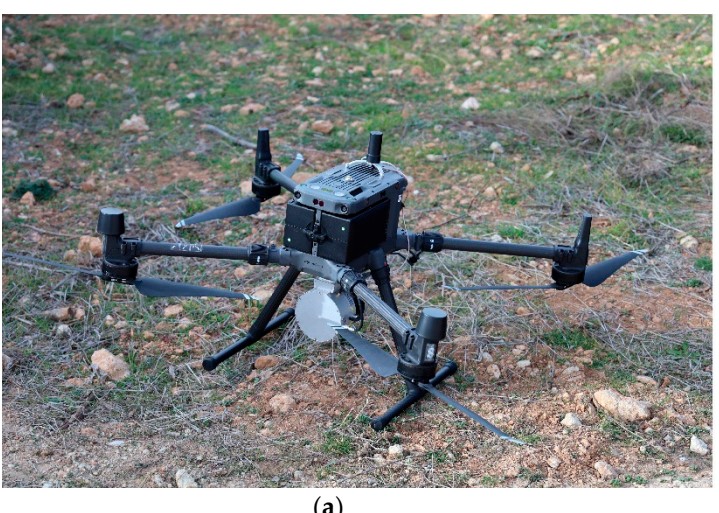

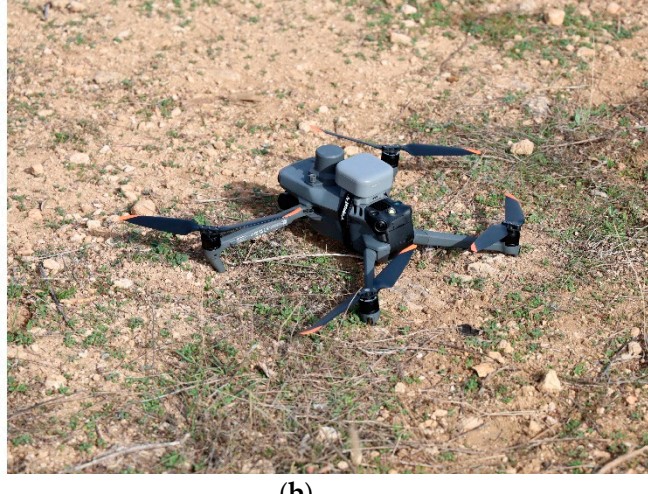

(**a**)  (**b**)

**Figure 9.** Drones used in flights: (**a**) ULIDAR board on DJI Matrice and (**b**) DJI Mavic 3 pro with parachute.

### *2.3. Photogrammetric System for UAS*

The system used to capture the photogrammetrical data was the DJI Mavic 3 pro RTK UAS (Figure 9b) equipped with a Hasselblad camera (Gothenburg, Sweden) that included a CMOS 4/3 sensor and 20 MP size; the equivalent focal length was 24 mm; the remaining characteristics could be seen in Table 2.

**Table 2.** Features of the DJI Mavic 3 pro RTK.

| Hasselblad Camera Boarding the Mavic 3 pro RTK | Features |
|---|---|
| Real Time Kinematic (RTK) module | GNSS capabilities: (GPS: L1C/A; L2C/L2P; BDS: B1I B2I; GLO: G1 G2; GAL: E1 E5b; QZSS: L1 L2) Fixed RTK uncertainty: Horizontal: 1 cm + 1 ppm; vertical: 1.5 cm + 1 ppm |
| Sensor size | CMOS 4/3 |
| Pixel size | 3.2 μm |
| Image size | 5280 × 3956 |
| Focal | 24 mm equivalent |
| Focus | 1 m a ∞ |
| ISO | 100–6400 |
| Shutter speed | 8-1/8000 s |
| Image formats | JPEG DNG (RAW) |

### *2.4. Methods*

To correctly identify the objects and structures in our scenario, it is very useful to have the colored point cloud. The Velodyne LiDAR alone does not produce such a cloud and to achieve it, additional photogrammetric flights have been carried out to color the complete LiDAR point cloud. The method used to obtain the colored point cloud is shown in Figure 10:

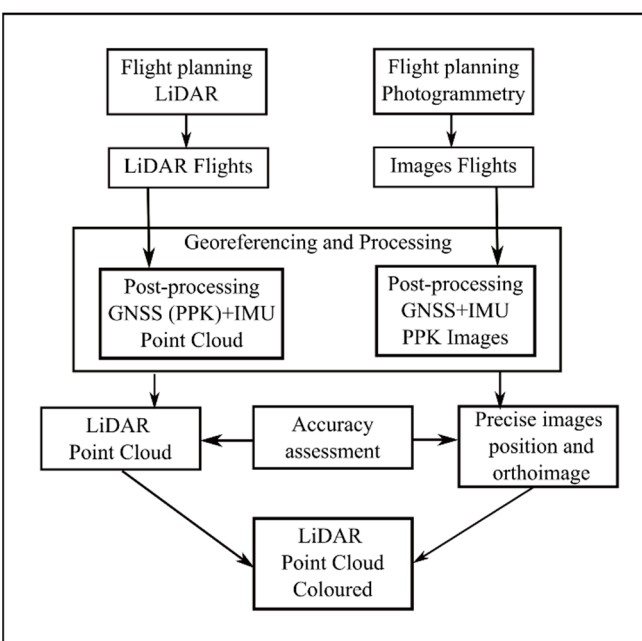

**Figure 10.** Workflow to obtain the colored point cloud.

2.4.1. Mission Planning

Before flying the drones, several reconnaissance visits to the site were necessary to plan both the most optimal time of year to carry out the flights and the strategic points from which to assemble all the equipment and direct them. Without a doubt and due to the existence of a large number of deciduous trees in the area, we considered late winter would be the best time to collect field data, between the fall of all the leaves that begins in autumn and before the first green shoots in spring.

It was also decided precisely due to the large amount of vegetation present in the place, LiDAR would be the best option to obtain a model in which the elements that were part of this spectacular cultural landscape were well-defined, as it would affect it to a lesser measure than to photogrammetry the density of said vegetation.

With the previous conditions of the orography and vegetation, we carried out an in situ tour of the entire area of the Los Tajos Natural Monument, on foot along its bottom and by car along its edges. We reach the following considerations:

- the study area has an area of 61 hectares (Figure 11), so it is necessary to plan the flights by dividing the area into several parts. Considering the duration of the flights in relation to the range of the battery, it is decided there is four points from which to carry out the flights and the Tajos is divided into five areas. In Figure 12, the 5 flights can be identified by the 5 pairs of circles that indicate the calibration prior to data collection for each of them.
- To capture field data from each of the strategic points, two consecutive flights will be executed: one with the DJI Matrice 300 drone, to which the Velodyne Puck Hi-Res LiDAR will be attached (Figures 13 and 14), and another with the DJI Mavic 3 pro drone that will take the photos that will be used to color the points that will be obtained with the LiDAR (Figure 15).

The LiDAR flight parameters included in the planning were as follows: flight altitude 60/70 m; to keep a constant altitude, the DTM05 (5 m resolution) from the National Geographic Institute was used in the flight plan; the different trajectories guaranteed a minimum cross-overlap of 50% and the flight speed was 7 m/s. An over-limitation of 20 m is established to guarantee the whole area of interest is covered.

The photogrammetric survey was composed of 2 flights oriented perpendicularly to each other (Figure 16). The flight parameters were as follows: flight height of 120 m

with a ground sample distance (GSD) of 3.3 cm; the longitudinal overlap was 80% and the transverse overlap 70% with a flight speed of 9 m/s and a safety over-limitation of 20 m at the perimeter of the area of interest. The number of flights planned was 8.

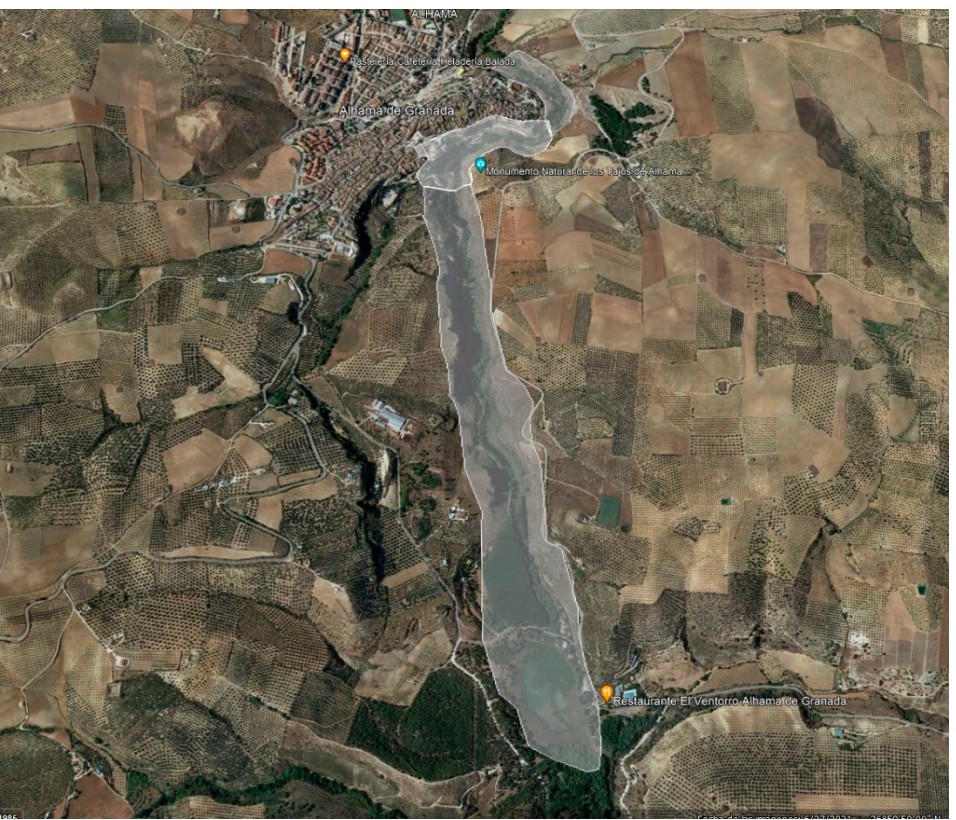

**Figure 11.** LiDAR and photogrammetric survey area (Pantaneta Lake and the Tajos).

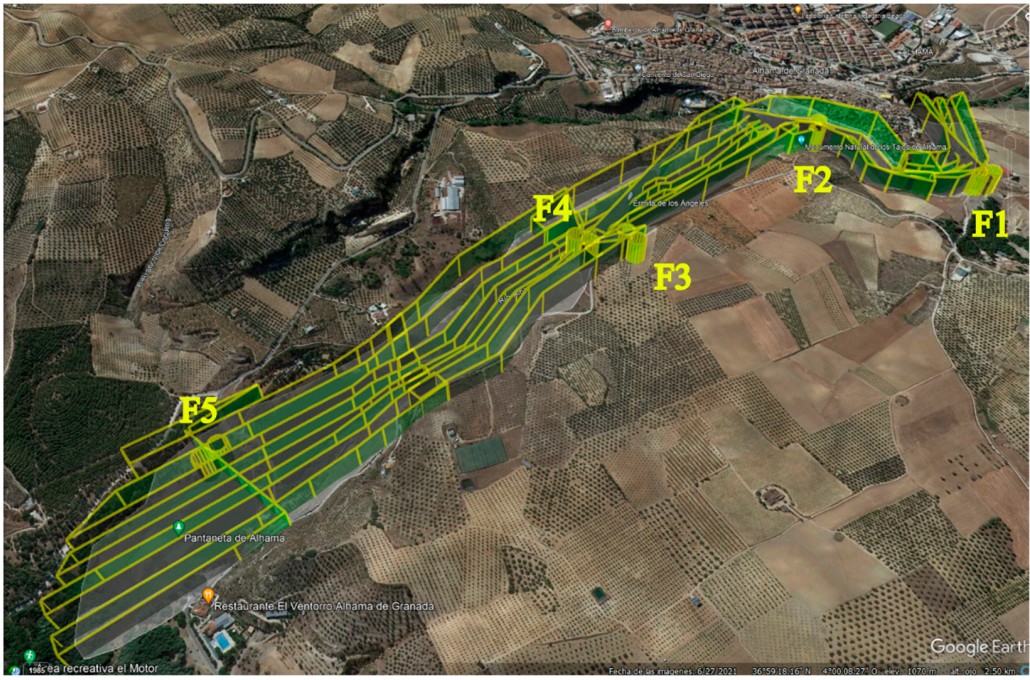

**Figure 12.** Planning of the 5 flight zones into which we divide the Tajos and the 4 points from which we direct the flights.

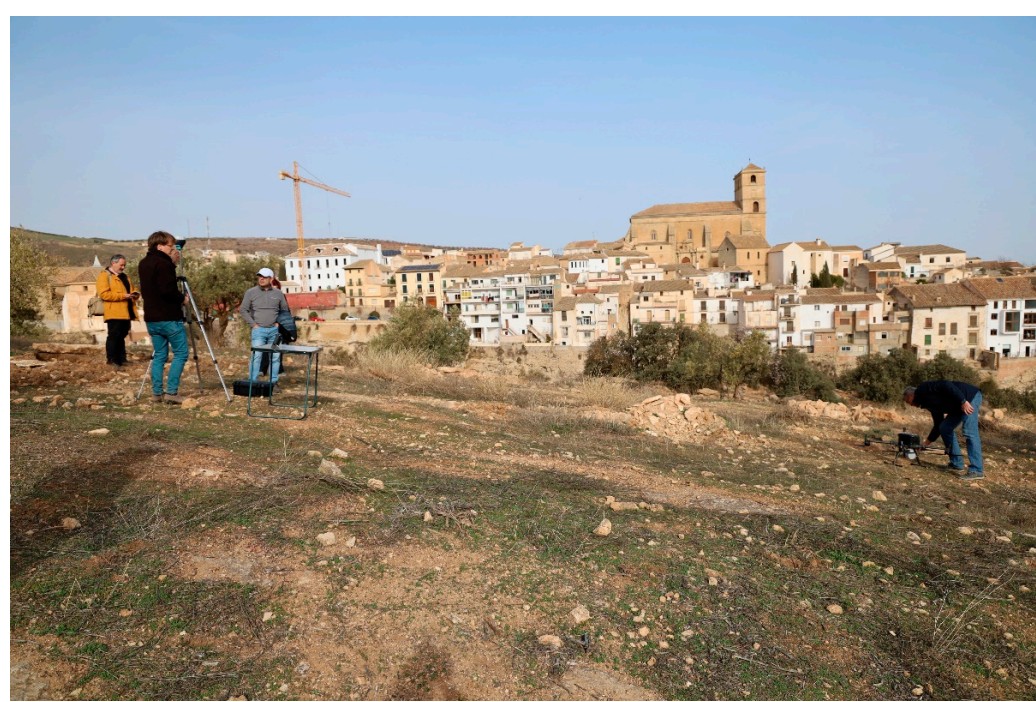

**Figure 13.** Equipment for the flight carried out with LiDAR.

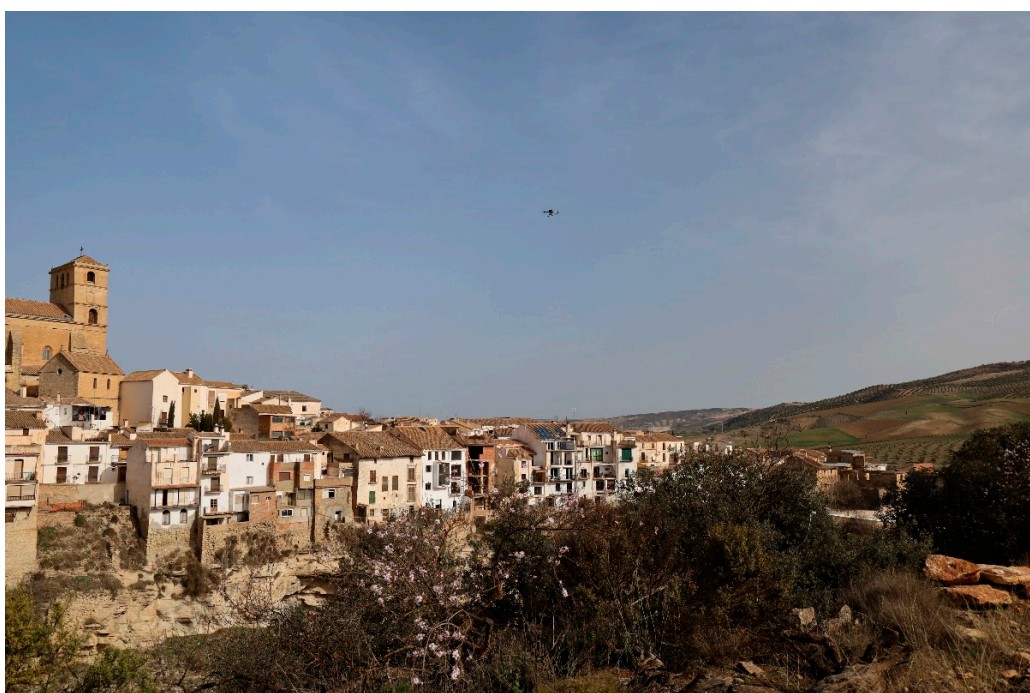

**Figure 14.** LiDAR flight over the Tajos.

The main LiDAR calibration task is focused on the IMU device because an accurate IMU data have the most influential strength on the resulting point cloud. The IMU calibration is performed at the starting flight, consisting of driving an 8 shape as soon the Matrice UAS achieves the flight attitude. This 8 shape calibration is prepared in the office by the software UgCS v.4.17 that is specialized in planning flights. The Figure 12 shows 5 IMU calibrations next to the F1, F2, F3, F4, and F5 texts.

The Mavic 3 Pro camera was previously calibrated to achieve a better result and to facilitate an auto-calibration computation in the post-processing handling. This will be exposed in the results section.

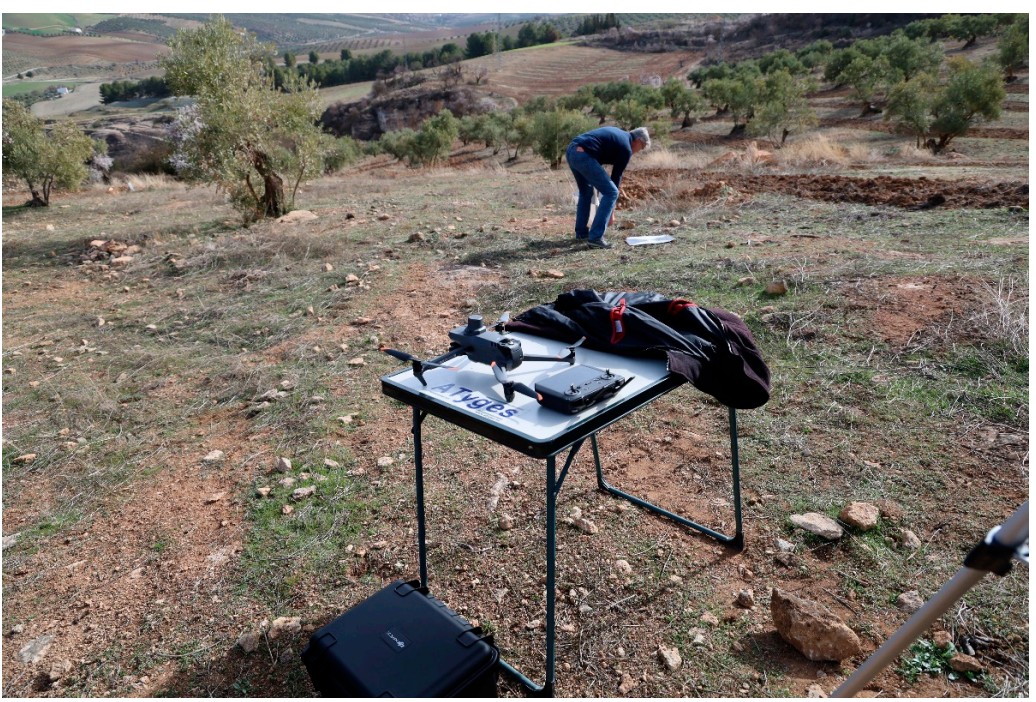

**Figure 15.** Flight preparation with DJI Mavic 3 pro.

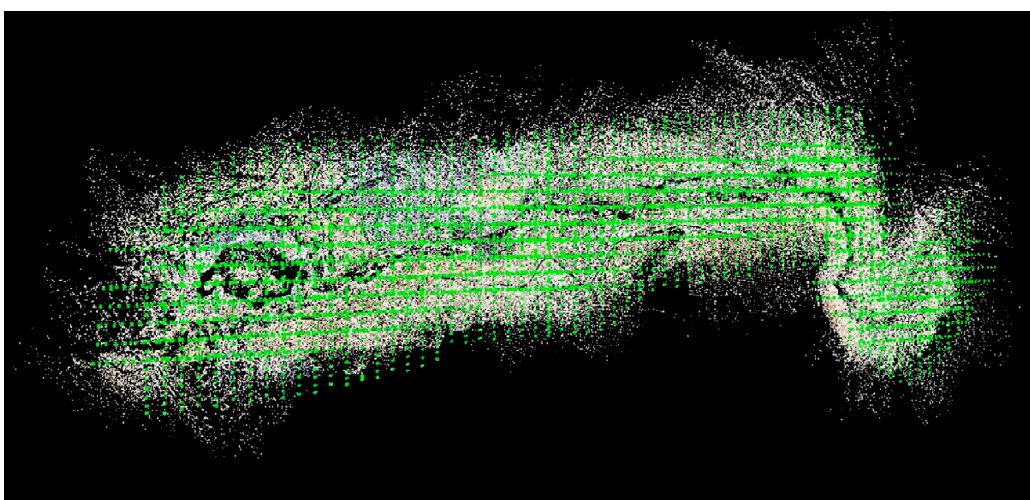

**Figure 16.** Perpendicular trajectories followed by the UAS on its 8 photogrammetric flights.

2.4.2. Georeferencing and Processing

LiDAR and photogrammetric flights are processed independently to obtain the colored point cloud. The results will be a LiDAR georeferenced point cloud and an orthophoto in the same reference system from the photogrammetric process. The reference system will be the official one in Spain, namely ETRS89 UTM zone 30.

To achieve georeferencing in the aforementioned reference system, the GNSS data were post-processed using PPK (post-processed kinematic) calculations. For this purpose, a GNSS base stationed in the vicinity of the flight areas were used to record GNSS data during the entire flight time. The coordinates of this station were computed using the RAP (Andalusian positioning network) as reference stations.

The software used to obtain accurate PPK coordinates to georeference the LiDAR on the Matrice 300 was Topodrone, the same software used to compute the coordinates of the individual points in the point cloud after transformation from the GNSS PPK data and the IMU orientations.

The software used to georeference the UAS Mavic 3 pro by means of PPK was Terra 3.7 from DJI; this software gave us different quality reports of the photogrammetric survey, both the orthophoto and the point cloud.

The precise PPK positioning of the cameras was supplied to Agysoft's Metashape software that, after importing the point cloud (processed with Topodrone), made it possible to color the point cloud.

To assess the products' accuracy, a series of control points were spread throughout the survey, and their coordinates were used to check the LiDAR survey uncertainty as well as the positioning of the photographic block needed to color the point cloud.

### 2.4.3. Point Cloud Product

The point cloud LiDAR is obtained by processing the flight using the Topodrone software v.1.0: the data GNSS and IMU from the UAS are used along with the UAS model, antenna height, and vector displacement of the LiDAR with respect to the antenna (Figure 17). To compute the PPK positioning, the base station data (antenna height, precise coordinates, and observation files) are introduced. All these data, and the IMU calibration data, are used to compute the position for each LiDAR return obtained during the flight and the result is saved in a ".laz" format.

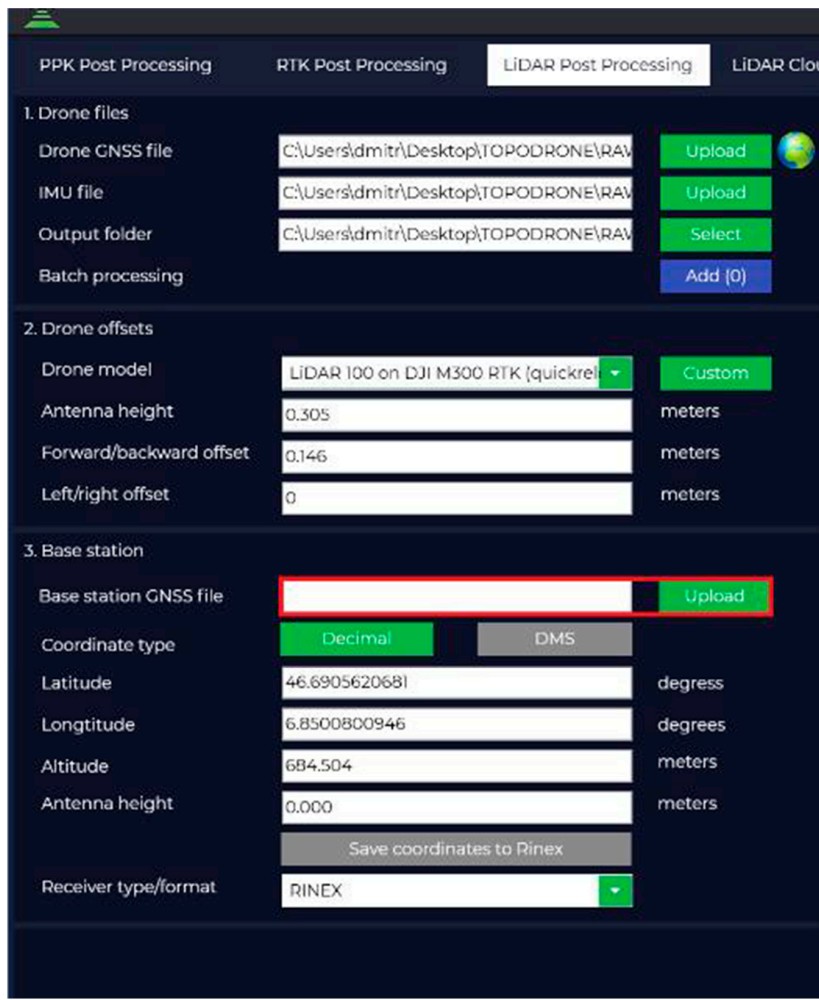

**Figure 17.** Post-processing: IMU, drone offsets, and GNNS data.

The point cloud obtained in the previous step lacked the true color it had in the real world. To color the point cloud, we proceed every point using the Metashape software. The photogrammetric block computed by Terra 3.7 is supplied to Metashape, together with the point cloud coming from Topodrone 1.0, so that, being both in the same reference system, it is possible to accurately color every point in the point cloud.

## 3. Results

After processing the LiDAR flight, we obtained a georeferenced point cloud in ETRS89 UTM zone 30 with 327 million points covering an area of 115.59 Ha with an average density of 283 pts/m2, a value much higher than the minimum we had set as objective (10 pts/m2). A density distribution diagram can be seen in Figure 18.

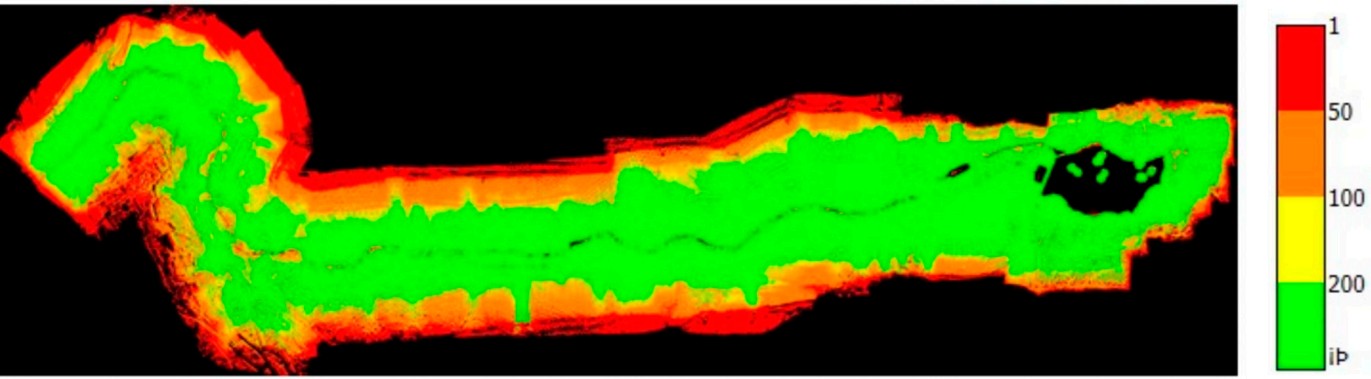

**Figure 18.** Density distribution from the georeferenced point cloud (pt/m2).

To make correct interpretations of the landscape elements, it was necessary to color the whole point cloud shown in Figure 19a. The average resolution produced by the point cloud density can be seen in Figure 19b, where a partial perspective from the surroundings of the flour mills located in the natural area of the Tajos de Alhama de Granada is shown. That perspective shows the trees that motivated the decision to carry out the survey by LiDAR rather than photogrammetry.

As explained in the methodology section, the LiDAR survey accuracy has been computed comparing the altitude coordinate between the control points (CPs) coming from GNSS and their homologous nearest points in the point cloud. Because the average density is 283 pts/m2, we achieve a good uncertainty estimation (Figure 19a shows the location of the GNSS-captured control points).

Table 3 shows the discrepancies between the control points and their homologous ones in the point cloud, from which the statistics that estimate the uncertainty in the point cloud georeferenced coordinates are computed. It shows the average altimetric error is 2.5 mm. However, it seems more suitable to use the absolute value for the mean error computation arises a mean error of 3.25 cm, which is still a very appropriate uncertainty value for our work objectives.

The photogrammetry to color the point cloud was generated by 4042 RTK georeferenced images captured by the Mavic 3 pro UAS operating at an average flight altitude of 114.58 m involving a 3.3 cm GDS. The photographs overlap distribution over the whole surveyed area could be seen in Figure 20, most of which was covered by more than 10 photographs.

As explained in the mission planning section, a previous camera calibration was performed, and after the computation alignment, an auto-calibration result was produced, so that the camera parameters were refined in this process. Table 4 shows the calibration camera parameters before (initial) and after (optimized) the flight.

The georeferencing root mean square error of this photogrammetric processing was 0.121 m. Taking advantage of the photogrammetric block structure, the orthophotography was extracted to facilitate the interpretation of objects not well identified in the point cloud. The orthophoto and a detail of it can be seen in Figure 21.

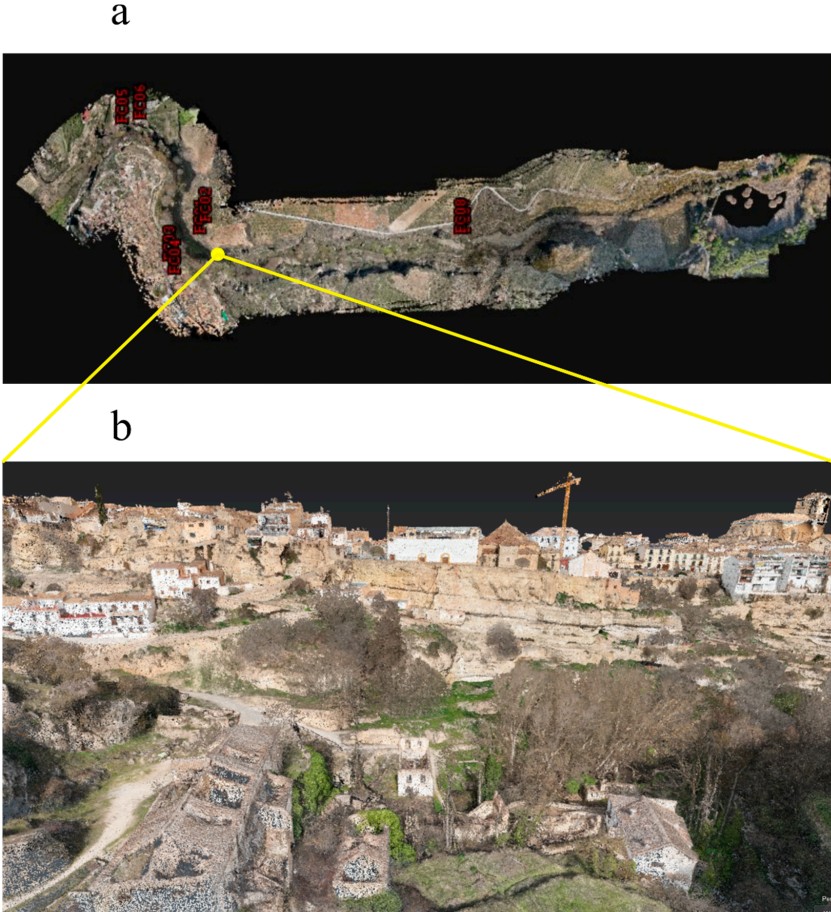

**Figure 19.** Point cloud: (**a**) LiDAR control points distribution, (**b**) detail from the colored point cloud.

**Table 3.** Point cloud quality control: differences between GNSS CPs and their homologous LiDAR CPs.

|  | GNSS | | | LiDAR | | |
|---|---|---|---|---|---|---|
|  | X (m) | Y (m) | Z (m) | Z (m) | $Z_{GNSS}$-$Z_{Lid}$ | ABS ($Z_{GNSS}$-$Z_{Lid}$) |
| PC01 | 412,289.911 | 4,095,554.729 | 886.029 | 885.996 | 0.033 | 0.033 |
| PC02 | 412,327.580 | 4,095,537.463 | 888.438 | 888.475 | −0.037 | 0.037 |
| PC03 | 412,199.919 | 4,095,657.578 | 882.833 | 882.745 | 0.088 | 0.088 |
| PC04 | 412,154.845 | 4,095,639.854 | 884.071 | 884.052 | 0.019 | 0.019 |
| PC05 | 412,657.949 | 4,095,809.973 | 850.159 | 850.189 | −0.03 | −0.03 |
| PC06 | 412,673.244 | 4,095,752.229 | 850.802 | 850.844 | −0.042 | 0.042 |
| PC07 | 412,289.298 | 4,094,665.816 | 946.680 | 946.691 | −0.011 | −0.011 |
| PC08 | 412,287.500 | 4,094,679.555 | 946.538 | 946.538 | 0 | 0 |
| Mean Error |  |  |  |  | 0.0025 | 0.0325 |

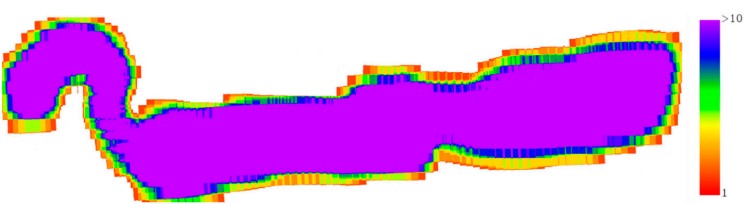

**Figure 20.** Photogrammetric overlap distribution: most surveyed area is covered by more than 10 photographs.

**Table 4.** Initial camera calibration and optimized settings after alignment.

| Parameter | Initial Values | Optimized Values | Difference |
|---|---|---|---|
| Focal (pixel) | 3713.29 | 3714.07 | 0.78 |
| Cx (pixel) | 2647.02 | 2641.059 | −5.961 |
| Cy (pixel) | 1969.28 | 1970.352 | 1.072 |
| K1 | −0.11257524 | −0.110056581 | 0.002518659 |
| K2 | 0.01487443 | 0.0083427 | −0.00653173 |
| K3 | −0.02706411 | −0.022595434 | 0.004468676 |
| P1 | −0.00008572 | −0.000008925 | 0.000076795 |
| P2 | 0.0000001 | −0.000296146 | −0.000296246 |

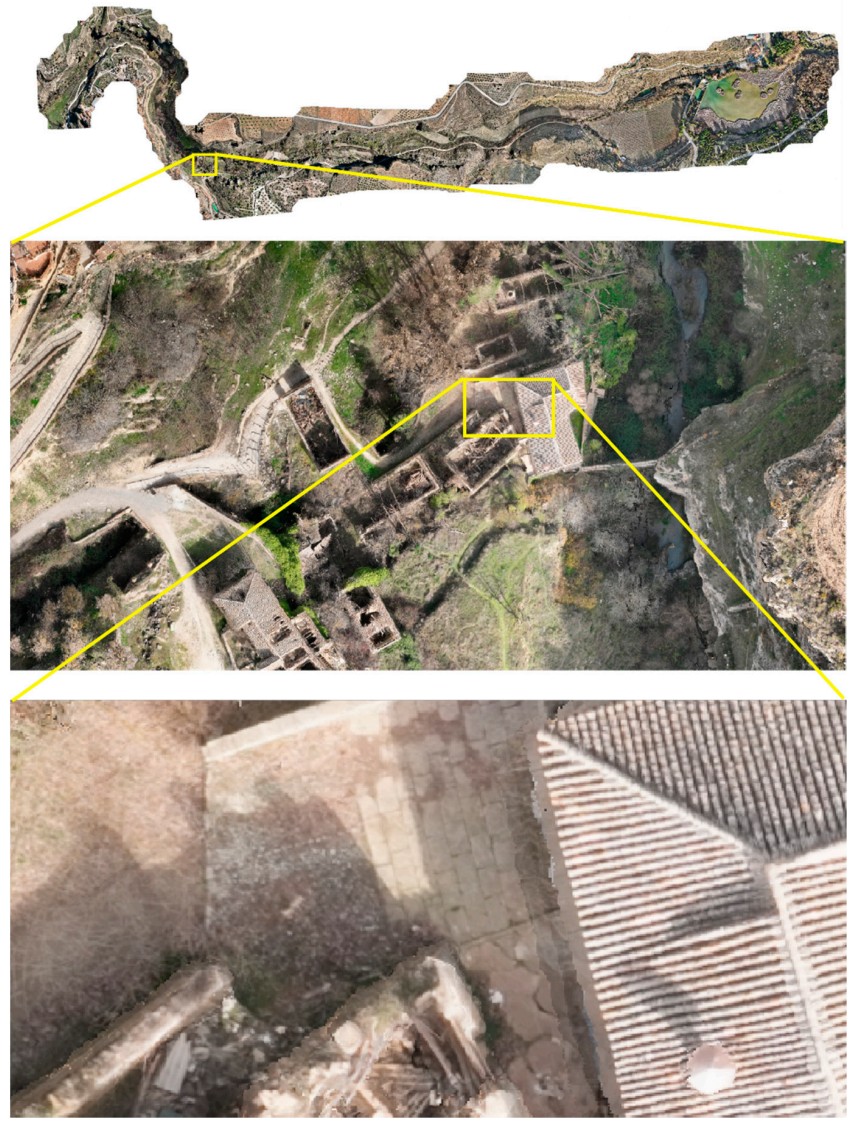

**Figure 21.** Orthophoto derived from the photogrammetry process and increasing details from a part small area.

## 4. Discussion

In this section we want to highlight the advantages we have achieved in our research by using a methodology based on a LiDAR system for UAS, which has resulted in a high-resolution 3D model of the heritage environment of the Tajos.

The main advantage that the use of LiDAR offered, compared to the models obtained from photogrammetry, was the penetration capacity that the laser had through the dense vegetation of the place, especially in some areas, obtaining a large number of points that would have been impossible in a photogrammetric survey. Figure 22 shows how in the river bed area, covered by trees, photogrammetry has problems to identifying homologous points and produces void areas (b), while LiDAR has more penetration and recovers ground returns under the trees, as well as very low herbaceous cover (a). This allows to have a more complete model to derive a digital terrain model after its classification.

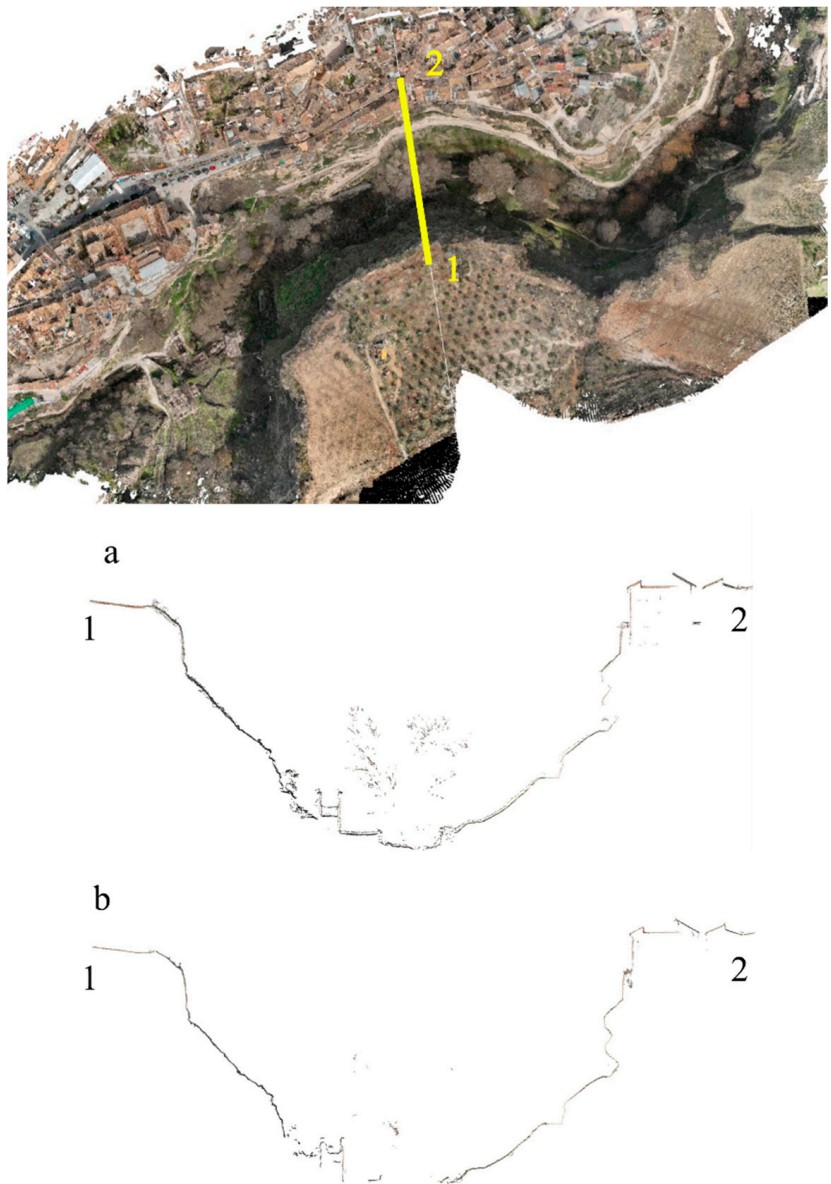

**Figure 22.** Cross-section from the point cloud LiDAR (**a**) and photogrammetry (**b**).

Figure 22 shows that slopes are very steep, becoming almost completely vertical in some sections. For this reason, the use of the LiDAR velodyne with 360° field of view capability facilitates the scanning task if it is horizontally mounted. This is achieved by reducing the number of trajectories as well as the need to record a larger number of overlapping scans that can introduce more noise and uncertainty.

LiDAR also offered other characteristics that, in our case, made it more suitable compared to the photogrammetric model:

- the ability to reconstruct fine elements that interfered with this heritage landscape, such as the high-tension electric cables that crossed the Tajos Natural Monument at various points.
- The high density of points that the LiDAR system was capable of offering, with great precision in terms of the coordinates generated in each point detected on the ground, compared to photogrammetric methods.
- The processing of the data obtained with the LiDAR required less time than any program to obtain the point cloud from the captured photographs, although we recognize this advantage did not apply in our case, since we had to do two flights, the LiDAR and the photogrammetric, to be able to color the LiDAR point cloud.

In the same way, it must be recognized the LiDAR system presented certain limitations or disadvantages compared to the photogrammetric model:

- the size of the drone needed to transport the LiDAR system was larger and heavier than drones with high-resolution cameras on the market.
- Working with LiDAR requires more sophisticated components and sensors, which adds complexity to the management of all field operations.
- These two disadvantages mentioned increased the cost of using a LiDAR system compared to a photogrammetric survey.
- The points obtained with LiDAR did not have the color that corresponded to them in reality as they did in photogrammetry.

The first three disadvantages were resolved by having ATyges, a company dedicated exclusively to the manufacture of drones and LiDAR and UAS technology systems, depending on the specialization of each job. They carried out the assembly of the LiDAR system and the flights, under the instructions of the research team of this article.

The last disadvantage, the absence of real color in the points, was taken into account from the beginning in our methodology and therefore it was decided to execute a second flight in each area with a commercial drone that incorporated a high-resolution camera to color the points obtained with the LiDAR from oblique photographs. The chosen drone was the DJI Mavic 3 pro.

The model obtained should serve, on the one hand, as a valuable tool for knowledge and analysis of all the elements (river, ditches, dams, mills, aqueducts, and paths) that made up this picturesque landscape (Figure 23), and on the other hand, as a basis for the future rehabilitation and architectural restoration projects that would have to be undertaken to preserve this cultural and landscape legacy.

In the conservation and preservation aspects, it is possible to use the detailed survey to detect the roofs and walls state of the mills to undertake restoration works, as well as identify places where ditches disappear so it is possible to retrieve them.

Part of the beauty of the Tajos de Alhama is their high rock gorges that enclose the site. Their high rock walls imply a hazard rockfall situation, and the LiDAR survey allows us to identify hazard point where intervention can be planned to reinforce the rock to the ground. This way, landscape preservation and people's safety will be achieved.

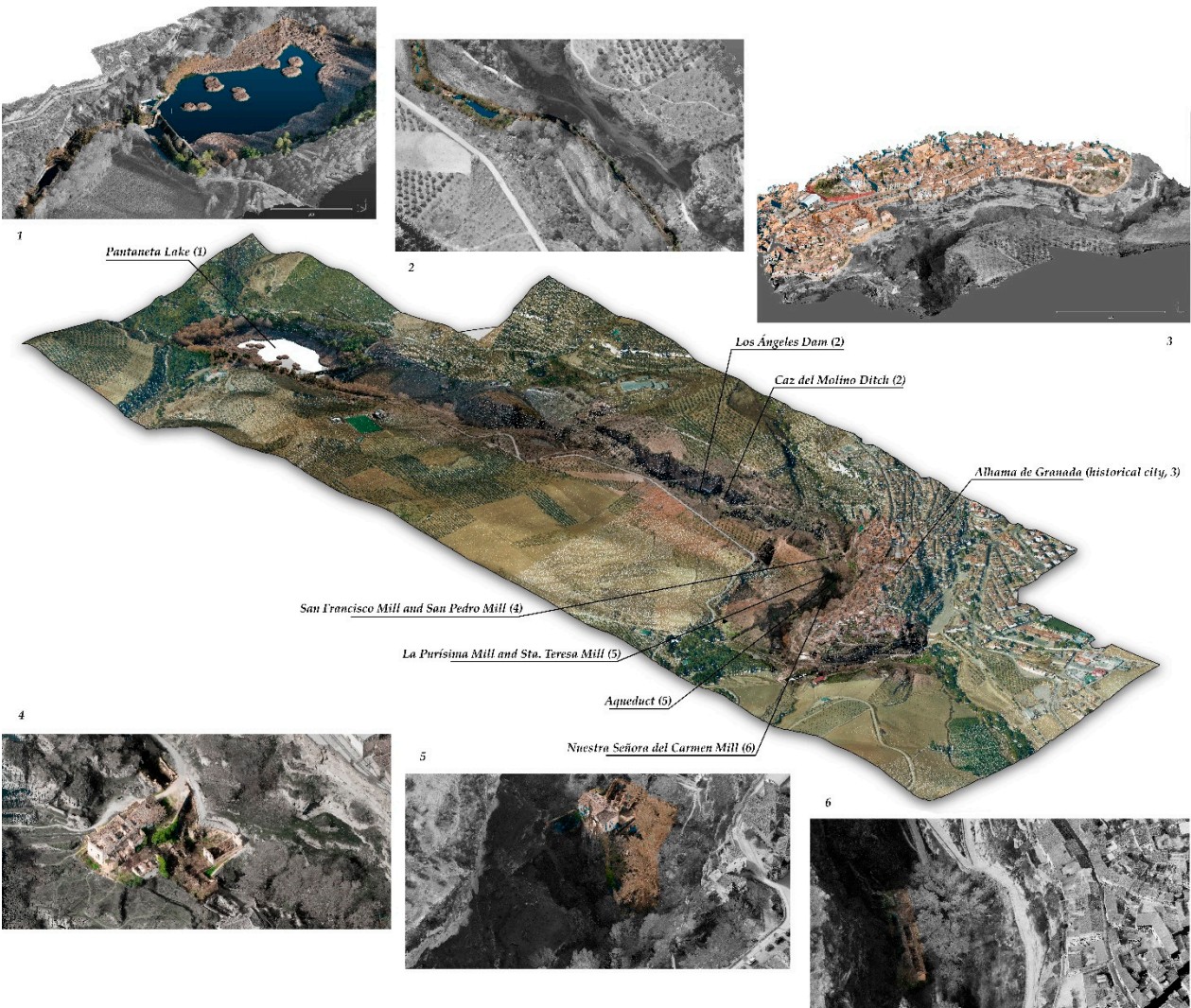

**Figure 23.** Point cloud LiDAR. Main elements of the Tajos Landscape, through which the Marchán River flows.

## 5. Conclusions

The Tajos de Alhama de Granada constitute a unique cultural heritage landscape of great beauty and historical importance, linked to its orography and the presence of water. In this research, we have demonstrated the remote origin of the city of Alhama bordered by the gorges, its presence in the writings of the Roman period and the maps that were made from this period from the Renaissance onwards, its Arab splendor and, already in the Christian era, the first drawings that we preserved of this landscape, from the 16th century. We have compiled different written and graphic testimonies from foreign travelers who visited the city since the 18th century and especially in the 19th century to verify everyone was impressed by this spectacular landscape.

Despite being currently classified at the highest level of protection, today it is in a progressive process of degradation, especially of its water flour mills. This research team has managed to carry out a survey, using a methodology that included the use of UAS LiDAR and photogrammetry, of the entire Tajos area. The results obtained showed a cloud of high-resolution 3D points in color, whose objective should be to serve as a basis for the subsequent identification of all the elements that made up this territorial enclave and the restoration projects that must be undertaken to preserve this cultural legacy.

In this research, we have shown the choice of a LiDAR, which also allowed the 360° rotation of the laser beams, on board the UAS has turned out to be the most appropriate due

to the existence of the large vertical walls of the Tajos and the presence of trees. At the same time, the need to achieve the color and materiality of the objects that made up this landscape, in the most realistic way possible, required coloring the point cloud obtained with the LiDAR, through a second flight with a commercial drone that had a high-resolution camera.

Therefore, the methodology followed that used a structured combination of UAS LiDAR technology and the photogrammetry offered by a commercial UAS has turned out to be the best option to obtain high-resolution, color 3D point cloud models of heritage landscapes, not very extensive with vegetation and even steep. The larger the dimensions of the area to survey, logically the greater the number of flights necessary. In our case study, five LiDAR flights and eight photogrammetric flights were necessary. If the area to be flown over was excessively large, we considered other options should be used such as airborne LiDAR on airplanes or helicopters, although undoubtedly in these cases the resolution of the point cloud obtained would be lower, since it would be necessary to fly at a higher altitude than in our case.

We plan to include the 3D heritage landscape model in a heritage building information model (HBIM) to improve the site management and its conservation and preservation, because we think a new layer including point cloud and building models are suitable for BIM technology.

**Author Contributions:** Conceptualization, M.d.C.V.-L. and J.F.R.-G.; methodology, M.d.C.V.-L., J.F.R.-G., J.G.M.-S., A.J.G.-B. and C.R.-M.; software, J.F.R.-G., M.d.C.V.-L. and C.R.-M.; validation, M.d.C.V.-L., J.F.R.-G., C.R.-M., A.J.G.-B. and J.G.M.-S.; formal analysis, M.d.C.V.-L., J.F.R.-G. and C.R.-M.; investigation, M.d.C.V.-L., J.F.R.-G., C.R.-M., A.J.G.-B. and J.G.M.-S.; resources, M.d.C.V.-L., J.F.R.-G. and C.R.-M.; data curation, J.F.R.-G. and M.d.C.V.-L.; writing—original draft preparation, M.d.C.V.-L. and J.F.R.-G.; writing—review and editing, M.d.C.V.-L. and J.F.R.-G.; visualization, M.d.C.V.-L. and J.F.R.-G.; supervision, M.d.C.V.-L., J.F.R.-G., C.R.-M., A.J.G.-B. and J.G.M.-S.; project administration, M.d.C.V.-L., J.F.R.-G., C.R.-M., A.J.G.-B. and J.G.M.-S.; funding acquisition, J.G.M.-S., M.d.C.V.-L., J.F.R.-G., C.R.-M. and A.J.G.-B. All authors have read and agreed to the published version of the manuscript.

**Funding:** This research was funded by the Junta de Andalucía (Spain), in a competitive call for Andalusian Public Universities for the development of research projects, under the jurisdiction of the Secretaría General de Vivienda, Project UGR.22-05 "DIETA-TICs. Documentación Integrada de Entornos paTrimoniales Andaluces con herramientas TIC's. Los Tajos de Alhama y sus molinos de agua".

**Data Availability Statement:** Data are contained within the article to this article.

**Acknowledgments:** Special thanks to Federico Alva and Ramón Martínez, from ATyges.

**Conflicts of Interest:** The authors declare no conflicts of interest.

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
