# Peer review of "High Resolution 3D Model of Heritage Landscapes Using UAS LiDAR: The Tajos de Alhama de Granada, Spain"

_land, doi:10.3390/land13010075_

Round 1

Reviewer 1 Report

Comments and Suggestions for Authors

I suggest justifying use of LIDAR in introduction. 

Comments on the Quality of English Language

I suggest clarifying singular vs plural noun and verb agreement. 

Author Response

- Q1. I suggest justifying use of LIDAR in introduction. 

- R1. We have added a justification in the introduction.

- Q2. I suggest clarifying singular vs plural noun and verb agreement. 

- R2. We have reviewed the manuscript and corrected or justified the agreement of nouns and verbs in singular or plural:

Line 11. The subject of the sentence is "The Tajos" not "landscape". The Tajos are several gorges.

Line 50. The Tajos are composed of several gorges in an unique valley.

Lines 60 to 63. These figures are categories of protection. Corrected.

Line 114. We are talking the Tajos: gorges (plural).

Line 225.  So that it is understood from the beginning of the article we have added an explanatory paragraph in the introduction.

Line 227. Corrected.

Line 234. Corrected.

Line 288. Corrected.

Line 310. Corrected.

Line 412. The Tajos are several gorges.

Reviewer 2 Report

Comments and Suggestions for Authors

One of the most attractive points of the proposal is to discuss how the character of that cultural landscape can be interpreted by the 3D model. That is if the layers of meaning included in the Introduction could be integrated into the model or an eventual digital twin. It is suggested to develop this discussion and to question what would differentiate the high-resolution 3D heritage landscape from any other territory without historical and cultural value.

Author Response

Following your suggestion, we have added a final paragraph in conclusions: "We plan to include the 3D heritage landscape model in a heritage building information model (HBIM) in order to improve the site management and its conservation and preservation, because we think a new layer including point cloud and building models are suitable for BIM technology."

Reviewer 3 Report

Comments and Suggestions for Authors

Great study. With minor additions, I believe it will be good for publication.

- Methodology Depth: Ensure the methodology section is sufficiently detailed, allowing other researchers to replicate the study. Consider adding more specifics about the LiDAR and photogrammetric systems, including the calibration and error handling processes.

- Technical Specifications: Provide more detailed specifications or models of the UAS and LiDAR systems used. Discuss the choice of technology and any limitations or advantages specific to the equipment.

- More discussion is needed on the broader implications of this research for heritage preservation, including how it can influence conservation strategies.

- Suggest directions for future research, such as applying the methodology to different types of heritage sites or integrating additional data layers for more comprehensive analysis.

Author Response

- Q1. Methodology Depth: Ensure the methodology section is sufficiently detailed, allowing other researchers to replicate the study. Consider adding more specifics about the LiDAR and photogrammetric systems, including the calibration and error handling processes.

- R1. At the end of Mission Planning section, it is introduced how the IMU calibration of LiDAR was performed as well as the initial camera calibration for the photogrammetry. In the results section we included the initial camera calibration parameters and the optimized after alignment in a new table (table 4).

- Q2. Technical Specifications: Provide more detailed specifications or models of the UAS and LiDAR systems used. Discuss the choice of technology and any limitations or advantages specific to the equipment.

- R2. Technical specifications have been added in tables 1 and 2.

- Q3. More discussion is needed on the broader implications of this research for heritage preservation, including how it can influence conservation strategies.

- R3. New discussions have been added to the end of Discussion section.

- Q4. Suggest directions for future research, such as applying the methodology to different types of heritage sites or integrating additional data layers for more comprehensive analysis.

- R4. New comments have been added at the end of conclusions.